# Applying Deep Learning on a Few EEG Electrodes during Resting State Reveals Depressive States: A Data Driven Study

**DOI:** 10.3390/brainsci12111506

**Published:** 2022-11-06

**Authors:** Damián Jan, Manuel de Vega, Joana López-Pigüi, Iván Padrón

**Affiliations:** 1Instituto Universitario de Neurociencia, Universidad de La Laguna, 38200 La Laguna, Santa Cruz de Tenerife, Spain; 2Department of Psychology, Faculty of Health Sciences, University of Hull, Kingston upon Hull HU6 7RX, UK; 3Departamento de Psicología Evolutiva y de la Educación, Campus de Guajara, Universidad de La Laguna, Apartado 456, 38200 La Laguna, Santa Cruz de Tenerife, Spain

**Keywords:** depression, long short-term memory, convolutional neural network, gated recurrent unit, non-linear features, diagnostic tool

## Abstract

The growing number of depressive people and the overload in primary care services make it necessary to identify depressive states with easily accessible biomarkers such as mobile electroencephalography (EEG). Some studies have addressed this issue by collecting and analyzing EEG resting state in a search of appropriate features and classification methods. Traditionally, EEG resting state classification methods for depression were mainly based on linear or a combination of linear and non-linear features. We hypothesize that participants with ongoing depressive states differ from controls in complex patterns of brain dynamics that can be captured in EEG resting state data, using only nonlinear measures on a few electrodes, making it possible to develop cheap and wearable devices that could be even monitored through smartphones. To validate such a perspective, a resting-state EEG study was conducted with 50 participants, half with depressive state (DEP) and half controls (CTL). A data-driven approach was applied to select the most appropriate time window and electrodes for the EEG analyses, as suggested by Giacometti, as well as the most efficient nonlinear features and classifiers, to distinguish between CTL and DEP participants. Nonlinear features showing temporo-spatial and spectral complexity were selected. The results confirmed that computing nonlinear features from a few selected electrodes in a 15 s time window are sufficient to classify DEP and CTL participants accurately. Finally, after training and testing internally the classifier, the trained machine was applied to EEG resting state data (CTL and DEP) from a publicly available database, validating the capacity of generalization of the classifier with data from different equipment, population, and environment obtaining an accuracy near 100%.

## 1. Introduction

Depression is a common mental disorder that comprises primary symptoms, such as depressed mood or anhedonia and loss of interest or pleasure, and secondary symptoms, such as appetite or weight changes, sleep difficulties (insomnia or hypersomnia), psychomotor agitation or retardation, fatigue or loss of energy, diminished ability to think or concentrate, feelings of worthlessness or excessive guilt and suicidality [1]. It is estimated that 3.8% of the global population is affected to some degree by depressive states at some moment of their lives, including 5% among adults and 5.7% of adults older than 60 years. Globally, 280 million people suffer from depression, and it is one of the critical/imperative conditions covered by WHO’s Mental Health Gap Action Programme (mhGAP) [2]. The Global Burden of Diseases, Injuries and Risk Factors Study showed that depression caused 34.1 million of the total years lived with disability (YLD), ranking as the fifth-largest cause of YLD [3].

Nowadays, depression diagnosis is mainly done through questionnaires and clinical interviews based on the expertise of professionals with the guidance of WHO’s International Classification of Diseases (ICD 11) or DSM-V in the USA. Such an approach does not provide an entirely reliable diagnosis, since the patient’s answers rely on subjective wellness or expectancy to improve their feelings, besides possible emotional links between patients and doctors. All of that can bias the interpretation of the questionnaires, which is necessary to advance in early depression detection through biomarkers that could help both in diagnosis and treatment evaluation [4,5].

The quest for objective diagnosis of mental disorders through non-invasive EEG has led to different approaches like the use of evoked potentials, band power, signal features, functional connectivity, and alpha asymmetry, some of them with mean accuracies of 90% [6]. The combination of resting state data and machine learning classification algorithms offers a promising new tool to improve the diagnosis and the evaluation of psychiatric disorders [7]. This combined method of resting state EEG and classificatory algorithms has also been applied to explore distinctive features of neural dynamics in depression [8]. Machine learning (ML) consists of computer algorithms that can automatically improve their performance by being trained with available data sets. A new area of ML involves deep learning (DL) algorithms, which combine multi-layers of the same or different basic architectural modules producing successful goals like speech recognition, translation, or even the creation of online chatbots that help people improve their mental health [9,10]. The growing field of applications based on robust deep learning has encouraged many researchers to use these methods with EEG resting state data classification [7], with special applications to explore psychopathological features and states, such as depression.

A recent review on the use of resting-state EEG data classification for depression [8] pointed out the principal pitfalls and successes of the research in this domain: (1) all of them used ML or statistical techniques while classifying; (2) non-linear features performed better; (3) electrodes were selected based on theoretical criteria of the researchers and ranged from 1 to 30 electrodes; (4) the accuracy showed ranged from 80 to 99.5%; (5) None reported a capacity for the generalization of their classifiers with data from other sources; (6) a lack of public EEG databases with resting state data on depressives is an obstacle. Recent studies overcome some of the proposed pitfalls. For instance, a large study [11] adopting a DL approach was validated with both a private database and an external database, using 16 channels. The model extracted spatiotemporal features through a convolutional neural network (CNN), combined with a gated recurrent unit (GRU), the former extracting spatial and frequency features and the latter capturing time sequence aspects of the extracted features. Their model reaches an accuracy of 89.63% in the classification of depression. A second study from the same laboratory [12], also using 16 channels, extended this solution by adding an attention layer and improving the results to 99.3%.

The current study aims to improve the classification of depression with nonlinear features, few electrodes, and standardized classifiers, as well as the model recently suggested by Liu et al. [11]. In this study, EEG resting state data were collected from a group of participants with depressive state and a group of matched controls, aiming to choose the best classification algorithm with a reduced number of electrodes and limited time-window, and their generalization was tested for an external sample. The design included four steps, first, selecting a space–time boundary of the time-window span and group of key electrodes, to use in further analyses. The second step was identifying the key non-linear features and appropriate classifiers that better capture the neural dynamics for the selected time-window and electrodes. The third step was tuning, training, and validating the selected classifier. The final model was trained with data collected in our laboratory, as described here. Fourth, the trained model was fed with external resting state data from a publicly available database, validating the generalization capacity of the obtained model and the possibility of the replication of our study.

In the first step, a logistic regression (LR) with a standard scaler (M = 0, SD= 1) was used as a baseline classifier. The use of basic non-linear features (Higuchi’s fractal dimension and sample entropy), suggested by Čukić et al. [8], was chosen to explore the optimal time window span and group of electrodes. For such a purpose, we used Giacometti’s algorithm that links Desikan’s brain structural areas with 10–20 standard EEG electrode distribution [13].

In the second step, after having selected the time window and electrodes with first-step results, different non-linear features were explored: Higuchi’s fractal dimension (Higuchi) is an approximate value for the box-counting dimension used on fractal analysis to graph a real-valued time series that has been used for more than 20 years in neurophysiological domains [14].Spectral entropy (SpectEnt) describes the complexity of a system based on applying the standard formula for entropy to the power spectral density of EEG data. It quantifies the irregularity of EEG data [15].Singular value decomposition entropy (SVDEnt) characterizes the information content or regularity of a signal depending on the number of vectors attributed to the process [16].Sample entropy (SampEnt) is a modification of approximate entropy, used for assessing the complexity of physiological time series signals [17].Detrended fluctuation analysis (DFA) is a stochastic process, chaos theory, and time-series analysis. DFA is a method for determining the statistical self-affinity of a signal. It is useful for analyzing time series that appear to be long memory processes [18].Permutation entropy (PermEnt) gives a quantification measure of the complexity of a dynamic system by capturing the order relations between values of a time series and extracting a probability distribution of the ordinal patterns [19].

In the second step, different classifiers were also tested: Logistic regression (LR): a baseline dichotomic classifier from machine learning (ML).Support vector machine (SVM): a robust ML algorithm that maps training examples to points in space (vectors), searching to maximize the distance between categories.Multi-layer perceptron (MLP): one of the simplest deep learning (DL) models used in supervised learning, consisting of fully connected layers.Convolutional neural network (CNN): another DL algorithm mainly used in image data analysis due to its biological visual architecture similarity.Long short-term memory (LSTM): a DL algorithm mainly used in sequential data analysis like natural language processing or time-series analysis.CNN + GRU (CNNGRU): a DL algorithm suggested by Liu et al. [11] that combines a CNN as a feature extractor plus a gated recurrent unit (GRU), an improvement of LSTM.

In the third step, after fixing the environments with the previous steps’ results, the selected model was tuned, trained, and validated with the data of our own experiment.

Finally, in the fourth step, the machine trained with our own dataset, resulting from such architecture and methodological approach, was tested with external data from a different lab, population, and environment to verify the generality of this approach.

## 2. Materials and Methods

This study protocol was approved by the Human Research Ethics Committee of the University of La Laguna, Tenerife, Spain, with number CEIBA 2021-3100, to protect the participants’ right according to the Declaration of Helsinki and was accepted by the board of the University Institute of Neuroscience (IUNE) from Universidad de La Laguna. 

### 2.1. Participants

A total of 50 participants (24 control and 26 depressive) were selected. The control group (mean age = 19.6, SD = 2.5), included 14 females and 10 males, 23 right-handed and 1 left-handed. The depressive group (mean age = 22.43, SD = 1.4), consisted of 23 females and 3 males, 22 right-handed and 4 left-handed. The two groups did not differ statistically in age: *t* (48) =1.28, *p* > 0.05). All participants had normal or corrected-to-normal vision and hearing.

To select the participants for the two groups, an online pre-screening was carried out through different faculties of the Universidad de la Laguna. The Beck depression inventory II (BDI, online version) [20] was chosen as the best measurement of the symptoms of depression. Control participants were selected among BDI’s lower than 13 and depressive participants with BDI scores higher than 20. The BDI consists of 21 items assessing the severity of symptoms. Minimal levels range from 0 to 13; this measure was used to select controls for our study. Mild depression ranges from 14 to 19 points, moderate 20–28, and severe higher than 29 points. As it was difficult to find students with severe depression for our study, we took as depressive anyone between moderate and severe depression. The control group (*n* = 24) had a mean BDI of 3.87 (SD = 2.32). The depressive group (*n* = 26) had a mean BDI of 32.92 (SD = 5.96). A *t*-test contrast between both groups gave *t* (48) = −22.32, *p* < 0.001.

### 2.2. Procedure

#### 2.2.1. First Step: Selecting the Time-Window Span and Key Electrodes

##### EEG-Resting State

We used a Neuroscan system with an Easycap of 64 electrodes to carry and acquire EEG data with a sampling rate of 500 Hz. Two electrodes placed up and below the left eye were used to keep control of blinks and vertical eye movements (VEO).

The participants were asked to remove jewelry and piercings. Then they signed an informed consent form and received instructions to ensure their understanding of the EEG procedure and task. The participants were asked to stay calm and keep their eyes on a fixation point while we recorded EEG for 3 min for the open-eyes resting-state condition. Afterwards, they were asked to close their eyes and stay as calm as possible for 3 min while we recorded the EEG closed-eyes resting-state condition (EC). Impedances were kept less than 10 kΩ during the whole experiment. For this analysis, we used only time segments of the EC condition, as it seemed to be the most accepted while classifying depression [6]. After this, the participants completed the Edinburgh handedness inventory [21].

##### Preprocesing

One participant of the DEP group was discarded due to data corruption. Therefore, the total number of participants submitted to the analyses were 49 (25 DEP and 24 CTL).

The collected individual EEG data were preprocessed automatically with MNE-Python [22] using the artifact subspace reconstruction (ASR) algorithm [23], implemented with Python’s “asrpy” package. An average reference was used, and a passband lower than 0.1 Hz and higher than 120 Hz was applied. Furthermore, a notch filteri of 50 Hz and a low-pass band filter attenuated frequencies above the cut-off (90 Hz), and with the filtered data, new files were created for each participant.

##### Time-Window Span Analysis

From each preprocessed participant file, a time window that ranged from 1 s to 17 s in gaps of 2 s was cropped after the trigger that identified the EC condition. These segments were resampled to 200 Hz to reduce processing time. An auxiliary miscellaneous channel we named ‘Y’ was added to the data structure to keep the participant characteristics using 0 to represent DEP and 1 to represent CTL.

All the segments were equalized on channels, concatenated and normalized through Normalize scalar with a L2 metric. The whole dataset was then split on vectors (x, y) of 200 measures each one (200 was the sampling frequency); *x* kept the voltage values of each electrode and *y* the type of participant (CTL or DEP). Higuchi’s fractal dimension (Higuchi) and sample entropy (SampEnt) features were calculated over all the electrodes of each (X, Y) vector, mapping them to a new (X, Y) pair where X contained the 2 non-linear features with mean values of all the electrodes and Y kept the type of participant.

A 10 K-fold cross validation test harness was applied through StratifiedKfold from the Scikit-learn python package [24] to a LR classifier (M = 0, SD = 1) with such (X, Y) vectors that were split randomly each time as X_train and X_test. The score results were kept to be analyzed later. This procedure was repeated for each of the time-window span segments (1, 3, 5, 7, 9, 11, 13, 15, 17), and their accuracy score results were kept.

##### Key Electrodes Identification

Once the time-window span was fixed, (x, y) vectors were constructed with each one of Desikan’s brain areas associated with electrodes [13]. Then such (x, y) vectors were processed to obtain new (X, Y) vectors keeping Higuchi and SampEnt on X and the type of participant in Y. The same procedure of 10 K-fold cross validation was applied, and the accuracy scores kept for later analysis. The entire procedure was repeated for each of the 32 electrodes, grouping the associated Desikan’s brain areas.

#### 2.2.2. Second Step: Exploring Features and Classifiers

In dynamic systems, entropy is associated with the rate of information production as a measure of the uncertainty linked to random variables, so the more information we have the less uncertainty we get. By measuring entropies on EEG data, when we seek accurate classification of complex dynamical processes, we are trying to reduce uncertainty. Some features built in this way can have shared knowledge, measured by mutual information. Therefore, the ranking of features through mutual information is a common way to reduce dimensionality when selecting features.

##### Features Extraction and Selection

A group of six non-linear features were extracted with the “antropy” python package, developed by Raphael Vallat [25]. The selected features were obtained from the time-window span of the selected electrodes provided by previous steps. A ranking feature was applied using mutual information, as this method is based on the measure of uncertainty reduction when comparing a target value of one variable against the feature assigned value; higher values represent a high connection between the feature and the target.

*SelectKBest* function from SKlearn was used. Once ranked, they were added one by one and retested while the accuracy of the LR classifier kept improving. Finally, six combinations were tested.

COMB0: PermEntCOMB1: PermEnt + SampEntCOMB2: PermEnt + SampEnt + SVDEntCOMB3: PermEnt + SampEnt + SVDEnt + DFACOMB4: PermEnt + SampEnt + SVDEnt + DFA + SpectEntCOMB5: PermEnt + SampEnt + SVDEnt + DFA + SpectEnt + Higuchi

The same procedure of K-fold test harness was applied as previously described keeping the scores for further analysis.

##### Testing Different Classifiers

Once fixed the features to be used, a comparison analysis between machine learning (logistic regression, support vector machine) and deep learning algorithms (multilayer perceptron, convolutional neural network, long short-term memory, CNNGRU) was performed, in order to check which approach gave the best classification scores. Machine learning approaches were based on Scikit-learn python packages while deep learning approaches were based on Keras python packages [26]. 

#### 2.2.3. Third Step: Tuning, Training, and Validating with the Experimental Data

##### Tuning and Training the Selected Model

The selected models were tuned and trained with the fixed parameters already explored on previous steps. The data were divided into 80% for training and 20% for validation. Once the models were fixed, its classification capacity was tested with the data collected at the lab. That means that the same files that were used to develop the models were then verified through the classifiers. 

#### 2.2.4. Fourth Step: Validating the Models with External Data

The success obtained with our own data could represent a situation wherein the models learned to classify some particular noise that correlated with the type of participant. Therefore, we applied the models to an external public database containing an EEG resting-state dataset of participants that were tested with the same BDI questionnaire as our experimental sample. The external sample was identified in PRED-CT’s EEG resting-state depression and controls data [27]. The selected external participants were CTL (*n* = 75, M = 1.73, SD = 1.65) and DEP (*n* = 30, M = 25.10, SD = 3.19); a t-contrast test between CTL and DEP gave *t* (183) = −49.12, *p* < 0.001. We selected DEP participants with scores higher than 25 in the BDI, since this was the cutting point for our experimental DEP participants with whom the classifier was trained.

The files of the external participants, with the same ranges of BDI scores, were preprocessed applying the same methods as described previously, which means that they were preprocessed automatically using the ASR algorithm, average-referenced, passband-filtered (0.1 Hz and 120 Hz), notch-filtered at 50 Hz and low-pass band filtered to attenuate the frequencies above the cut-off (90 Hz), then a time window of 15 s was cropped from each file after the trigger that identified the EC condition. These segments were resampled to 200 Hz, and an auxiliary miscellaneous channel named ‘Y’ was added to keep the class of the associated group (CTL = 1, DEP = 0); in addition to keeping the three electrodes selected (AFz, FC2, F2), the resulting (X, Y) vectors were fed to the locally trained models to test its performance through accuracy, precision, recall, and F1 scores. 

## 3. Results

### 3.1. First Step: Exploring the Time-Window Span and Key Electrodes

#### 3.1.1. Time-Window Span Analysis

As shown in Figure 1, a time-window span of 15 s was appropriate to keep enough data from each participant file when using the LR classifier and Higuchi and SampEnt as basic non-linear features. Paired *t*-tests were applied around the selected 15 s point, and pair *t*-tests between 15 s and 9 s were so significatively different that they were not included. 

15 s vs. 17 s: *t*-test independent samples, *t* (49) = −0.206, *p* = 0.83813 s vs. 15 s: *t*-test independent samples, *t* (49) = −0.211, *p* = 0.83411 s vs. 15 s: *t*-test independent samples, *t* (49) = −1.695, *p* = 0.1079 s vs. 15 s: *t*-test independent samples, *t* (49) = −7.152, *p* < 0.0001

#### 3.1.2. Identifying Key Electrodes

To detect which group of electrodes were more accurate in classifying controls/depressive participants, we applied an exploratory loop following Desikan’s brain structural areas mapped on associated electrodes as shown in Giacometti et al. [13]. Fifteen seconds of the closed-eyes resting state were cropped from each participant file and processed by extracting two non-linear features (Higuchi and SampEnt) to feed a five-fold cross-validation logistic regression (LR) classification. This was systematically applied on the associated regions’ electrodes as an exploratory baseline (see Table 1).

Statistical comparisons with a paired *t*-test were applied to the validation scores of the four selected brain areas: the lateral orbitofrontal cortex (LOFC), caudal anterior cingulate cortex (CACC), precuneus (PREC), and superior frontal gyrus (SFG). The results, illustrated in Figure 2, showed significant differences between CACC and PREC (*t* (48) = 49.25, *p* < 0.0001) and between CACC and SFG (*t* (48) = 31.49, *p* < 0.0001) but not between CACC and LOFC (*t* (48) = 0.028, *p* = 0.977). Therefore, the caudal anterior-cingulate area and the associated electrodes (FC2, AFz, F2) with high mean accuracy (M= 0.987, SD = 0.0006) were kept for further analyses. 

### 3.2. Second Step: Exploring Features and Classifiers

#### 3.2.1. Features Extraction and Selection

Figure 3 shows the distributions of non-linear features: permutation entropy (PerEnt), sample entropy (SampEnt), single-value decomposition entropy (SVDEnt), detrended fluctuation analysis (DFA), spectral entropy (SpectEnt), and Higuchi’s fractal dimension (Higuchi) by groups (CLT on the left and DEP on the right) after applying a MinMaxscaler.

According to their distributional values, the ranking of features was calculated using mutual information before testing their capacity to classify the groups. Figure 4 shows a comparison of its importance.

Next, six combinations of non-linear features were tested and compared, beginning with the one with highest importance and adding a new feature according to their rank at each combination. Below are the combinations:COMB0: Permutation entropy (PerEnt)COMB1: PerEnt + sampling entropy (SampEnt)COMB2: PerEnt + SampEnt + single-value decomposition entropy (SVDEnt)COMB3: PerEnt + SampEnt + SVDEnt + detrended fluctuation analysis (DFA)COMB4: PerEnt + SampEnt + SVDEnt + DFA + spectral entropy (SpectEnt)COMB5: PerEnt + SampEnt + SVDEnt + DFA + SpectEnt + Higuchi’s fractal dimension (Higuchi)

To assess the relative accuracy of the combinations, a series of *t*-tests were performed between the obtained observations. The comparison, illustrated in Figure 5, indicates the superior accuracy of COMB5, which is significant with respect to COMB0, COMB1, COMB2, (*p* < 0.001), and non-significant with respect to COMB3, COMB4 (*p* > 0.05).

#### 3.2.2. Testing and Comparing Different Classifiers

Having fixed the time-window span to 15 s, the brain area CACC with its associated electrodes (FC2, AFz, F2) and the nonlinear features (PerEnt, SampEnt, SVDEntr, DFA, SpectEnt, Higuchi), a 10-fold cross-validation test harness to ML and DL classifiers was applied: logistic regression (LR), support vector machine (SVM) with radial basis function (RBF) kernel, multilayer perceptron (MLP), convolutional neural networks (CNN), long short-term memory (LSTM), and newest CNN + GRU (CNNGRU) proposed classifier by Liu W. [11]. The accuracy was selected as a metric score. The *t*-test comparisons between pairs of classifiers in accuracy led to the results illustrated in Figure 6. As can be seen, the highest accuracy was obtained for the LSTM and the CNNGRU, which did not differ significantly. However, the LSTM showed better performance than CNN (*p* <0.01) or LR, MLP, and SVP (*p* <0.0001). CNNGRU showed better performance than CNN (*p* < 0.0001) and LR, MLP, and SVP (*p* < 0.0001). 

LSTM and CNNGRU outperformed the other classifiers, so we decided to keep both; the first one represents a more classical approach, and CNNGRU is a promising approach. 

### 3.3. Third Step: Tuning, Training, and Validating with the Experimental Data

The tuned LSTM model consisted of four stacked LSTM layers with a Batch normalization layer previous to the final dense layer with sigmoid activation and kernel regularizer L2. The tuned CNNGRU model had a convolutional 1 D layer, a MaxPoling layer, a GRU layer, then a flattened layer prior to the final dense layer with sigmoid activation. Both models used binary cross entropy as the loss function, Adam optimizer, one hundred training epochs, and a batch size of 32. 

A check pointer was used to save the best weights to reconstruct the models for predictions without the necessity of retraining them. The data processed with previous results was split, with 80% for training and 20% for validation purposes. The graph of the loss function values as the training/validating progress during training epochs (Figure 7) is used to detect overfitting and underfitting. In underfitting cases, the validation curve is above the training one, while in overfitting, the validation curve, after matching the training in some point, starts to grow up, while the training curve remains low. In Figure 7, we see that, in the LSTM case (left), the validation curve converges to training more accurately after 80 training epochs, while in the CNNGRU model (right), both curves match each other, so neither over- nor underfitting affects the trained machines.

Once the trained models were kept fixed, the experimental files were preprocessed following the selected processing context analyzed through previous steps and fed to the classifier in order to detect the goodness of the obtained success. Figure 8 (left) shows the receiver operating curve (ROC) of the LSTM model after applying the validation test and (right) the ROC curve of the CNNGRU model with the same data; in both cases, the local classification gave 100% accuracy with an area under the curve (AUC) of 1.

### 3.4. Fourth Step: Validating the Model with External Data

The success of the models in classifying our experimental participants in step three with an area under the curve (AUC) of 1.0 could suggest that the model learned some artificial noise that correlated with our environment and/or participants. For this reason, we were compelled to test the trained models with the aforementioned external resting-state database.

To validate this approach, we downloaded a database from PRED-CT, the EEG resting-state depression and controls data [27]. Figure 9 (left) shows the ROC and AUC obtained after processing and feeding the external data into the LSTM model and (right) the same with CNNGRU model. It confirmed our hypothesis that a four-step methodological design focusing on only non-linear features, few electrodes, and a short time-window span is enough to obtain a good classifier that could help in detecting a depression state. 

Models obtained a high performance both in the classification of data obtained in the laboratory and in the classification of data that originated from a public database. Error-related measures were first obtained. That is, true positive (TP), true negative (TN), false positive (FP), and false negative (FN). Next, performance measures were computed as follow: Accuracy = (TP + T)/(TP + TN + FP + FN); Precision = TP/(TP + FP); Recall = TP/(TP + FN), and F1 = 2 × (Precision × Recall)/(Precision + Recall). The performance values for the two selected models are shown in Table 2.

## 4. Discussion

This study demonstrated that applying a deep-learning-based classifier with just three electrodes during 15 s of close-eyed EEG resting-state data, collected from a sample of college students, allows them to be accurately classified as depressed or non-depressed individuals. Once the classifier was internally trained, it successfully classified depressive and control participants with an accuracy of 100%. Moreover, it was successfully applied to detect depression in untrained EEG resting state data from a publicly available source, with a comparable accuracy of 99%. 

To achieve this remarkable performance, a data-driven methodological approach, consisting of training and comparing various machine learning and deep learning classifiers, using a selection of only three electrodes, a 15-s epoch, and a combination of six non-linear features. It turned out that the LSTM model and the new CNNGRU model were the most accurate. Both the LSTM and CNNGRU models retain a memory of the previous biosignal history, so they are especially appropriate in the analysis of time series wherein the hypotheses are related to time-extended signals, as is the case of the EEG resting state [27]. Although our study was data-driven, the fact that LSTM and CNNGRU were the best classifiers for depression suggests that the use of non-linear features in deep learning methods captures the long-standing and complex non-linear dynamics of the brain, serving as a potential diagnostic tool to detect abnormal EEG signatures associated with depression. 

Although the performance of both models on the same data is quite similar, the CNNGRU model is in some respects better, since it is less complex, having one less layer than the LSTM model, and also requiring a shorter training time than LSTM. Nevertheless, overall, the method developed in this article offers higher accuracy than the CNNGRU reported by Liu et al. [11], mainly due to the specific selection of features and electrodes. They needed 16 electrodes to classify depressive people with an accuracy of 89.63%, while the current methodology reached almost perfect accuracy with a selection of only three electrodes and a few non-linear features. These methodological choices are crucial, as the CNNGRU model also improved its performance with our own data, using our selected electrodes, features, and procedures.

The group of three electrodes selected in this study (FC2, AFz, F2) have been associated with the activity of the caudal anterior cingulate cortex, according to Desikan’s brain structural areas mapped on Giacometti’s electrodes mapping algorithm [13]. This is congruent with the literature reporting that the symptoms of depression are associated with anatomical and functional impairments in the caudal anterior cingulate cortex [28,29]. Note, however, that the anatomical parcellation methods and the assignment of groups of electrodes to brain regions are primarily descriptive. To provide more accurate neuroanatomical and functional data on depression, neuroimaging methods or source estimation algorithms for high-density EEG set-ups are necessary. The minimal approach of three-electrodes EEG, used in this study, is appropriate for searching for efficient machine learning classification algorithms but not for accurate source estimation.

With our data-driven method, we found that the choice of an EEG epoch of 15 s to train the algorithm was appropriate to detect depressive states. This fits well with the proposal that the measures of functional connectivity and network organization of the brain tend to stabilize around 12 s [30]. On the other hand, we can assume the non-linear features employed in our classificatory model are especially sensitive to events occurring in large time windows, which might index functional connectivity. For this reason, techniques with high temporal resolution, such as EEG and MEG, are potentially useful to capture the signatures of brain dynamics associated with depression and other mental health conditions [31].

However, it is possible that efficient classifiers of structurally based brain diseases require a combination of linear features [32], for instance, delayed cerebral ischemia and seizures after subarachnoid hemorrhage [33], while classifiers for functionally based diseases (like depressive states) rely on a combination of non-linear features [8]. From this point of view, we could think that depression seems to be a complex functional disease that cannot be accurately identified through simple linear features of the EEG. 

This study has some limitations, which may be addressed in future research. It was conducted with a population of young university students, half of whom showed depressive states, according to a self-report questionnaire, although none of them had been diagnosed with depression, and the results cannot be generalizable to other populations and depressive conditions. However, given the simplicity of the EEG resting-state protocol and the efficiency of the LSTM and CNNGRU as classificatory tools reported herein, the current methodology could be applied in the future to broader and more heterogeneous samples of population, including groups of different ages and clinically diagnosed patients. The classificatory algorithms and methods developed herein provide objective measures of brain dynamics, which can be used as a complementary diagnosis tool in clinics. It could be trained to distinguish among different types of depression, depression levels, and even anxiety and/or other comorbid disorders. In addition, it could be used for following up, as an objective guidance of treatment results, applying the tool to depression patients before and after their treatment to test their improvement, beyond subjective impressions. Finally, this study raises the possibility of creating a portable EEG device with three electrodes that is easy to use and even linked to smartphones, which could be used on primary care services to screen depressive cases, facilitating prevention.

## 5. Conclusions

This study demonstrates that, using only non-linear features and a few electrodes, we can train an algorithm to accurately classify young participants as depressed or non-depressed. Furthermore, the trained classifiers generalized their performance to external untrained databases. Deep learning approaches outperformed machine learning ones, as they obtained better classification values. Classifiers based on EEG data with few electrodes, like the ones used in this study, could potentially be implemented as brain computer interfaces (BCI) to be easily employed as a complementary tool to support diagnosis.

## Figures and Tables

**Figure 1 brainsci-12-01506-f001:**
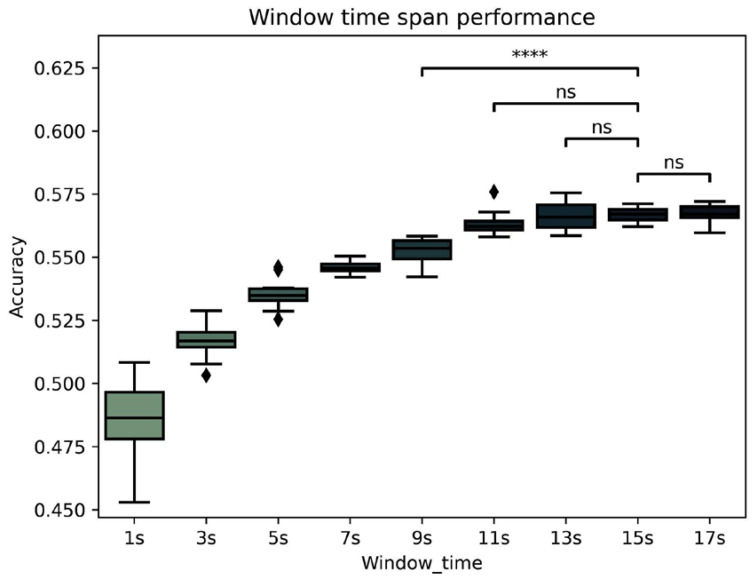
Classificatory accuracy comparison between time-window span. *p*-value annotation legend: ns: non-significant; **** *p* < 0.0001; diamonds points represent outliers.

**Figure 2 brainsci-12-01506-f002:**
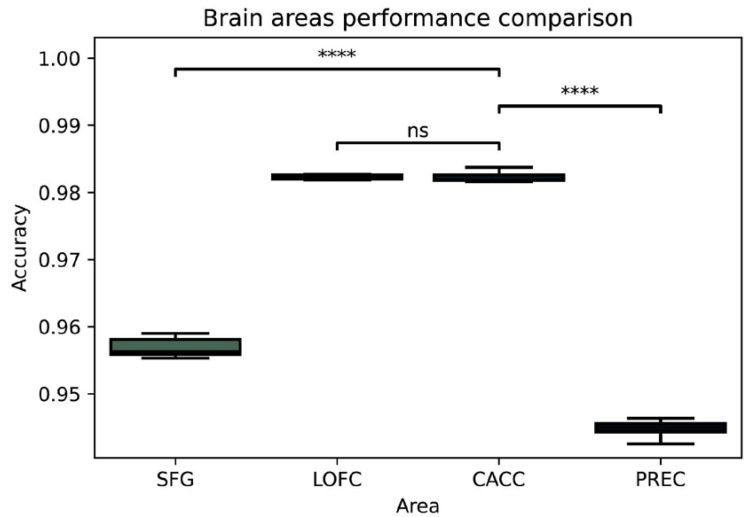
Accuracy comparison between brain areas associated with the electrodes at the lateral orbitofrontal cortex (LOFC), caudal anterior cingulate cortex (CACC), precuneus (PREC), and superior frontal gyrus (SFG). *p*-value annotation legend: ns: non-significant; **** *p* < 0.0001.

**Figure 3 brainsci-12-01506-f003:**
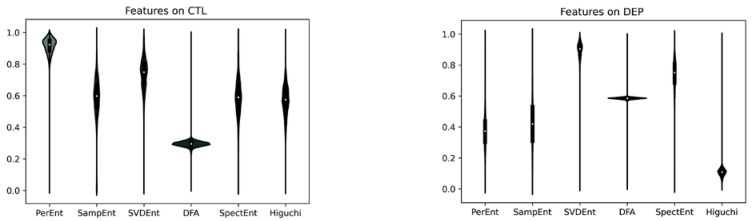
Distributions of values of non-linear features. Permutation entropy (PerEnt), sample entropy (SampEnt), single-value decomposition entropy (SVDEnt), detrended fluctuation analysis (DFA), spectral entropy (SpectEnt), and Higuchi’s fractal dimension (Higuchi). **Left**: CTL. **Right**: DEP.

**Figure 4 brainsci-12-01506-f004:**
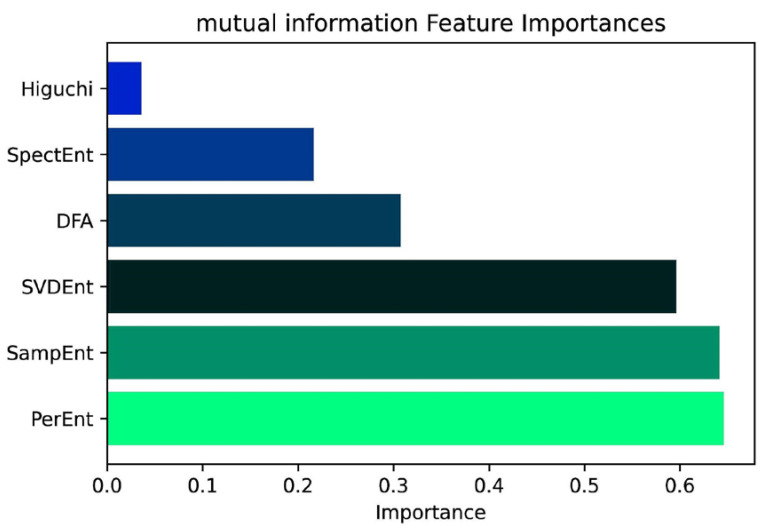
Features ranking through the mutual information of fermutation entropy (PerEnt), sample entropy (SampEnt), single-value decomposition entropy (SVDEnt), detrended fluctuation analysis (DFA), spectral entropy (SpectEnt), and Higuchi’s fractal dimension (Higuchi).

**Figure 5 brainsci-12-01506-f005:**
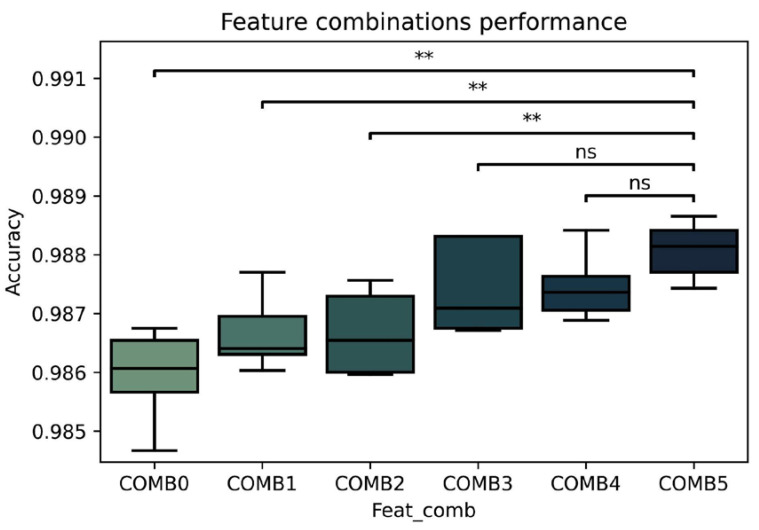
Accuracy comparison between non-linear selected features. COMB0 (PerEnt), COMB1 (PerEnt + SampEnt), COMB2 (PerEnt + SampEnt + SVDEnt) COMB3 (PerEnt + SampEnt + SVDEnt + DFA) COMB4 (PerEnt + SampEnt + SVDEnt + DFA + SpectEnt), COMB5 (PerEnt + SampEnt + SVDEnt + DFA + SpectEnt + Higuchi); *p*-value annotation legend: ns: non-significant; ** *p* < 0.01.

**Figure 6 brainsci-12-01506-f006:**
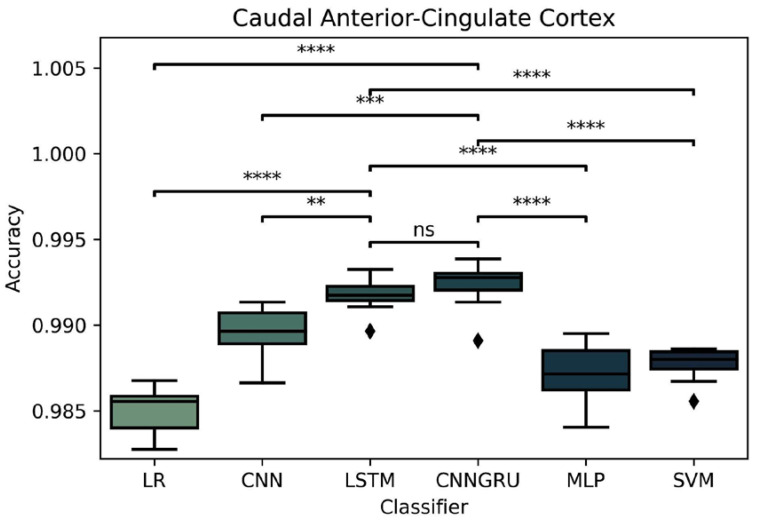
Comparison of the classifiers logistic regression (LR), convolutional neural network (CNN), long short-term memory (LSTM), CNN+ gated recurrent unit (CNNGRU), multilayer perceptron (MLP) and support vector machine (SVM); *p*-value annotation legend: ns: nonsignificant; ** *p* < 0.01, *** *p*< 0.001, **** *p* < 0.0001. Diamonds points represent outliers.

**Figure 7 brainsci-12-01506-f007:**
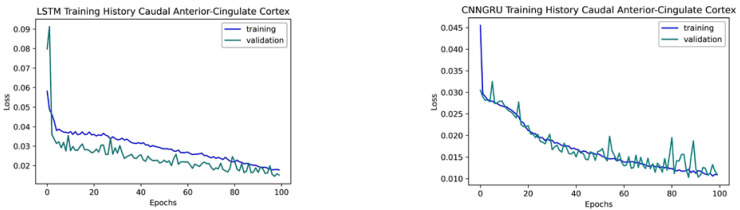
Training and validating the history of the models. **Left**: LSTM. **Right**: CNNGRU.

**Figure 8 brainsci-12-01506-f008:**
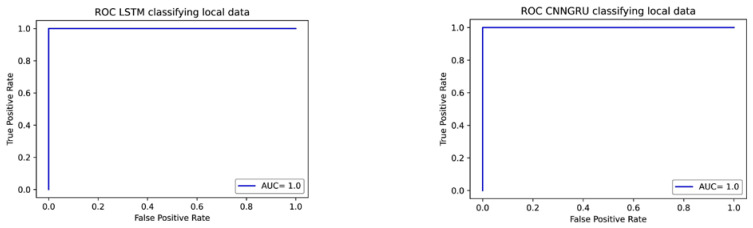
ROC and AUC curves from classifying local obtained data. **Left**: with the LSTM model. **Right**: with the CNNGRU model.

**Figure 9 brainsci-12-01506-f009:**
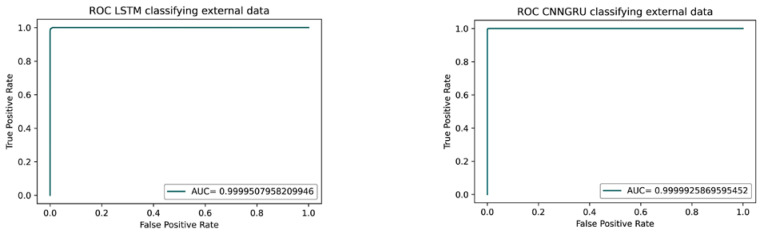
ROC and AUC curves while classifying external data from a public database. **Left**: LSTM model. **Right**: CNNGRU.

**Table 1 brainsci-12-01506-t001:** Associated regions and electrodes as an exploratory baseline. In bold, the regions and the electrodes with the highest validation score.

Region	Electrodes	Cross-Validation Score
Transverse temporal	TP8, T7, T8	0.660 (±0.01)
Banks superior temporal sulcus	P8, TP7, TP8	0.735 (±0.01)
Caudal anterior-cingulate	FC2, AFz, F2	0.987 (±0.0006)
Caudal middle frontal	FC6, FC3, FC4	0.791 (+/−0.003)
Isthmus-cingulate	PO8, PO7, Pz, POz	0.807 (+/−0.01)
Lateral occipital	PO7, PO8, O1, O2	0.853 (±0.0007)
Lateral orbitofrontal	N2, F8, N1, F9, F10	0.987 (±0.0006)
Lingual gyrus	Iz, PO8, PO7, Oz	0. 784 (±0.002)
Medial orbital frontal	Fp2, N2, Fpz	0.928 (±0.001)
Paracentral lobule	C1, C2, Cpz, Cz	0.931 (+/−0.001)
Pars opercularis	FC5, FC6, FT7, FT8	0.684 (±0.003)
Pars orbitalis	F8, AF7, F9, F10	0.918 (±0.001)
Pars triangularis	FT7, FT8, F5, F8, F7	0.685 (±0.004)
Pericalcarine	POz, O1, O2, Oz	0.908 (±0.0005)
Postcentral gyrus	T8, C5, C4, C6, C3	0.610 (±0.002)
Posterior cingulate	Cz, C1, FC2, C2	0.881 (±0.001)
Precentral gyrus	C3, C2, C4, C1	0.821 (±0.001)
Precuneus	PO3, Pz, POz	0.951 (±0.0009)
Rostral anterior cingulate	AF4, AFz, Fp2, Fpz	0.853 (±0.001)
Rostral middle frontal gyrus	AF3, F5, AF8, F6	0.740 (±0.002)
Superior frontal gyrus	AF3, F5, AF8, F6	0.961 (±0.0004)
Superior parietal	POz, CP1, P2, P1	0.894 (±0.0007)
Superior temporal gyrus	TP8, FT9, T8, T7	0.660 (±0.001)
Supramarginal gyrus	C5, CP5, CP6	0.816 (±0.001)
Cuneus	PO3, PO4, Oz, POz	0.919 (±0.0009)
Inferior parietal	P3, P4, P5, P6	0.842 (±0.002)

**Table 2 brainsci-12-01506-t002:** Comparative performance of LSTM and CNNGRU models.

Model	Loss	Accuracy	Precision	Recall	F1
LSTM	0.0157	0.99489	0.99246	0.99715	0.99480
CNNGRU	0.0118	0.99628	0.99651	0.99588	0.99620

## Data Availability

In this link you can check both databases. https://drive.google.com/file/d/1WmnRPaYu-Bm6pSgJjETBH7vBtHyADLMl/view?usp=sharing (accessed on 26 October 2022).

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
