# Peer review of "Applying Deep Learning on a Few EEG Electrodes during Resting State Reveals Depressive States: A Data Driven Study"

_brainsci, 2022, doi:10.3390/brainsci12111506_

Round 1

Reviewer 1 Report

The subject is interesting. The paper might be more significant if the authors had provided a more extensive discussion of the various techniques. However, certain aspects of the paper's technicality and presentation prevent me from accepting it, and I am marking it as the "Major Revision."

1-    Conclusions: What is stated in the conclusion section is not fully supported by the data shown in the paper.

2-    The authors extracted some types of features ( HFD, SE, SVDE, SPCTE) and then, put them together in different combinations. Why did the authors not put all extracted features beside together and use the rank-feature-based feature selection methods? The feature selection and combinations method is not efficient. Please compare the system's performance with a more efficient feature selection method (for example, rank-feature). For more details, check the following article: Automated detection of driver fatigue from electroencephalography through wavelet-based connectivity

3What is their main advantage over other methods? The authors recorded a good dataset, but the analysis part of the paper is not new.

4The comparative study from the recently proposed method is missing. It might be better to compare the performance and computational cost. More recently proposed methods should be reviewed and used for comprehensive comparisons.

5- The complexity of the proposed model and the model parameters' uncertainty are not mentioned. Please compare the complexity and running time with the previous articles.

6- The author needs to explain how the method helps the current field and what challenges it solves. Please elaborate more on the proposed approach with more focus on the relations between its components, as they are the core of the solution and need more justification for using them.

7- Please explain more about the validation method; it might be better to explain it in a separate section.

8- The literature review in the article seems insufficient. Similar approaches should be included in the introduction section or given as a separate section.

9- The results and discussions section should be reorganized in a more highlighting, argumentative way. I strongly recommend adding a comparison with some recent studies.

10- The research findings and contribution need to be stated clearly. As well as the obtained results in this paper. So, the authors are requested to connect the main idea and contribution with the obtained results to prove that the proposed method accomplished its main aims and better results.

11- The logic of the introduction can be improved. For example, the reasons and significance of applying deep/machine learning methods to the neurobiology study of disorders could be introduced. Current progress and critical issues could also be mentioned. The authors can use these articles to edit this section. A Depression Diagnosis Method Based on the Hybrid Neural Network and Attention Mechanism-- Computer aided diagnosis system using deep convolutional neural networks for ADHD subtypes - A Depression Prediction Algorithm Based on Spatiotemporal Feature of EEG Signal- Detection of child depression using machine learning methods

 12- Please use the violin plot to show the distribution of extracted features in the two class of data.

Author Response

REVIEWER 1

Comments and Suggestions for Authors

The subject is interesting. The paper might be more significant if the authors had provided a more extensive discussion of the various techniques. However, certain aspects of the paper's technicality and presentation prevent me from accepting it, and I am marking it as the "Major Revision."

RESPONSE: Thank you for your revision and your insightful comments and suggestions.

1-    Conclusions: What is stated in the conclusion section is not fully supported by the data shown in the paper.

RESPONSE: Thank you for your remark. We changed the conclusion as follows:

“This study demonstrates that using only non-linear features and a few electrodes, we can train an algorithm to accurately classify young participants as depressed or non-depressed. Furthermore, the trained classifiers generalized their performance to external untrained databases. Deep Learning approaches outperform Machine Learning ones as they get better classification values. Classifiers based on EEG data with few electrodes, like the ones used in this study, could potentially be implemented as Brain Computer Interfaces (BCI) to be easily employed as a complementary tool to support diagnosis.”

2-    The authors extracted some types of features ( HFD, SE, SVDE, SPCTE) and then, put them together in different combinations. Why did the authors not put all extracted features beside together and use the rank-feature-based feature selection methods? The feature selection and combinations method is not efficient. Please compare the system's performance with a more efficient feature selection method (for example, rank-feature). For more details, check the following article: Automated detection of driver fatigue from electroencephalography through wavelet-based connectivity

RESPONSE:  Thank you for your relevant suggestion. We increased the number of nonlinear features to 6: PermEnt + SampEnt + SVDEnt + DFA + SpectEnt + Higuchi; and ranked and combining them through mutual information (see new Figure 4), and kept them all to improve the methodology (new Figure 5, compares the accuracy of feature combinations). This point is clarified in the manuscript:

“A group of 6 non-linear features were extracted with “entropy” python package, developed by Raphael Vallat [24]. The selected features were obtained from the time window span of selected electrodes  provided by previous steps. A  ranking feature was applied using mutual information as this method is based on the measure of uncertainty reduction when comparing a target  value of one variable against the feature assigned value, higher values represent a high connection between the feature and the target. SelectKBest function from SKlearn was used. Once ranked they were added one by one and retested while the accuracy  of LR  classifier kept improving. Finally, 6 combinations were tested. “

3- What is their main advantage over other methods? The authors recorded a good dataset, but the analysis part of the paper is not new.

RESPONSE: Thank you for your suggestion, which allowed us to make explicit comparisons between our method and the method reported by recently published studies, and to enrich our discussion. We explained now in the discussion the outcome of these comparison and the main advantages of our method:

“Although the performance of both models on the same data is quite similar, the CNNGRU model is in some respects better, since it is less complex, having one less layer than the LSTM model, and also requiring a shorter training time than LSTM. Nevertheless, overall, the method developed in this article offers higher accuracy than the CNNGRU reported by Liu W. [10], mainly due to the specific selection of features and electrodes. They needed 16 electrodes to classify depressive people with an accuracy of 89,63%, while the current methodology reached almost perfect accuracy with a selection of only three electrodes and a few non-linear features. These methodological choices are crucial, as the CNNGRU model also improved its performance with our own data, using our selected  electrodes, features  and procedures.”

4- The comparative study from the recently proposed method is missing. It might be better to compare the performance and computational cost. More recently proposed methods should be reviewed and used for comprehensive comparisons.

RESPONSE: Following your important suggestion, we incorporated some of the proposed papers in the introduction and discussion, since they provide great insights to improve our paper. In the introduction we added this paragraph:

“Recent studies overcome some of the proposed pitfalls. For instance,  a large study [10] adopting a DL approach was validated with both a private database and an external database, using 16 channels. The model extracted Spatiotemporal features through a Convolutional Neural Network  (CNN), combined with a Gated Recurrent Unit (GRU), the former extracting spatial and frequency features, and the latter capturing time sequence aspects of the extracted features. Their model reaches an accuracy of 89.63% in the classification of depression.  A second study from the same laboratory [11], also using 16 channels,  extended  this solution by adding an attention layer and improving the results to 99,3%.”

RESPONSE: In addition, we decided to test the latest CNNGRU model used by Liu W et al 2022 with our own data. The model performed similar to the LSTM model, our previous best selection.  Nevertheless, as stated in our response to point 3, the key factor and difference between their approach and ours relies on shrinking the number of electrodes and feeding the models with different data, in the CNNGRU case with PSD’s  from 16  electrodes on 1  to 3 seconds slicing, and in our case with nonlinear features extracted from 15 seconds slicing windows. We obtained with our data and with external data higher accuracy than theirs using also a private and a public database.  Now, in the introduction we announced the inclusion of the CNNGRU model:

“The current study aims to improve the classification of depression with nonlinear features, few electrodes and standardized classifiers, as well as the model recently suggested by Liu, et al. [11].”

RESPONSE: In the Method section, we added the comparative performance of the new CNNGRU model, and the remaining algorithms tested in the manuscript. The description of the new algorithm is included in the manuscript as:

“CNN+GRU (CNNGRU): this DL algorithm suggested by Liu W. [11] combines a CNN as a features extractor plus a Gated Recurrent Unit (GRU), an improvement of LSTM. “

RESPONSE: Finally, the results incorporating the CNNGRU model, are reported in the updated Figure 6, in the new Table 2 and in the following paragraph:

“The accuracy was selected as a metric score. The t-test comparisons between pairs of classifiers in accuracy lead to the results illustrated in Figure 6. As can be seen, the highest accuracy was obtained for the LSTM and the CNNGRU, which did not differ significantly. However, LSTM showed better performance than CNN (p < .01), and LR, MLP and SVP (p < . 0001 ). CNNGRU showed better performance than CNN (p < .001), and LR, MLP, and SVP (p < .0001).”

5- The complexity of the proposed model and the model parameters' uncertainty are not mentioned. Please compare the complexity and running time with the previous articles.

RESPONSE: The comparison between the different models is updated including the new CNNGRU model (see also response to point 4). Even though CNNGRU reduces complexity and runtime in comparison to LSTM, the performance of the two models is equivalent and we decided to keep both in the revised manuscript, for the sake of comparison.  In addition, we emphasize in several places throughout the manuscript that an efficient classification of depressive states depends not only on the algorithm, but also on other methodological aspect such as the number of electrodes and the choice of features, which in the current manuscript are optimized.

6- The author needs to explain how the method helps the current field and what challenges it solves. Please elaborate more on the proposed approach with more focus on the relations between its components, as they are the core of the solution and need more justification for using them.

RESPONSE: The applied method led to a good solution for classifying depression through a data driven procedure coupled with suggestions from previous researchers. Thus, our method points out the importance of nonlinear features and the selection of electrodes by groups of representative brain areas as suggested by Giacometti’s algorithm. The method did not necessarily lead to the most efficient solution, but it did lead to a good practical solution by reducing the number of electrodes and increasing the classification accuracy. These ideas are developed at several places throughout the manuscript:

In the Abstract:

“ We hypothesize that participants with ongoing depressive states differ from controls in complex patterns of brain dynamics that can be captured in EEG resting state data, using only nonlinear measures on a few electrodes,  making it possible to develop cheap and wearable devices that could be even monitorized through smartphones.

In the section Method, 2.2.2

In dynamic systems, entropy is associated with the rate of information production as a measure of the uncertainty linked to random variables, so the more information we have the less uncertainty we get. By measuring entropies on EEG data,  when we seek accurate classification of complex dynamical processes, we are trying to reduce uncertainty. Some features built in this way can have shared knowledge, measured by  mutual information. Therefore, the ranking of features through mutual information is a common way to reduce dimensionality when selecting features.

7- Please explain more about the validation method; it might be better to explain it in a separate section.

RESPONSE: Explained now in the Methods:

“The files of the external participants, with the same ranges of BDI scores, were preprocessed applying the same methods as described previously, that means preprocessed automatically using ASR algorithm, average referenced, passband filtered (0.1 Hz and 120 Hz), notch filtered at 50 Hz and a low-pass band filtering to attenuate the frequencies above cut-off (90 Hz), then a time window of 15s was cropped from each file after the trigger thad identified the EC condition. These segments were resampled to 200Hz an auxiliar miscellaneous channel named ‘Y’ was added to keep the class of the associated group CTL=1, DEP=0, besides of  keeping the 3 selected AFz, FC2, F2; the resulting (X,Y) vectors were fed to the locally trained models to test it performance through Accuracy, Precision, Recall and F1 scores. “

8- The literature review in the article seems insufficient. Similar approaches should be included in the introduction section or given as a separate section.

RESPONSE: Thank you for your suggestion to include recent and relevant references. We added in the introduction:

“Recent studies overcome some of the proposed pitfalls. For instance,  a large study [11] adopting a DL approach was validated with both a private database and an external database, using 16 channels. The model extracted Spatiotemporal features through a Convolutional Neural Network  (CNN), combined with a Gated Recurrent Unit (GRU), the former extracting spatial and frequency features, and the latter capturing time sequence aspects of the extracted features. Their model reaches an accuracy of 89.63% in the classification of depression.  A second study from the same laboratory [12], also using 16 channels,  extended  this solution by adding an attention layer and improving the results to 99,3%. 

“The current study aims to improve the classification of depression with nonlinear features, few electrodes and standardized classifiers, as well as the model recently suggested by Liu, et al. [11]”

9- The results and discussions section should be reorganized in a more highlighting, argumentative way. I strongly recommend adding a comparison with some recent studies.

RESPONSE: Following the reviewer’s suggestion, the discussion has been entirely rewritten, adding new and expanding previous arguments, such as the following paragraphs :

“To achieve this remarkable performance, a data-driven methodological approach, consisting of training and comparing various machine learning and deep learning classifiers, using a selection of only three electrodes, a 15-seconds epoch, and the combination of six non-linear features. It turned out that the LSTM model and the new CNNGRU model were the most accurate. Both the LSTM and CNNGRU models  keep memory of the previous biosignal history, so they are especially appropriate in the analysis of time series where the hypotheses are related to time-extended signals, as is the case of the EEG resting state [27]. Although our study was data-driven, the fact that LSTM and CNNGRU  were the best classifier for depression suggests that the use of non-linear features in deep learning methods captures the long-standing and complex non-linear dynamics of the brain, serving as a potential diagnostic tool to detect abnormal EEG signatures associated with depression.”

“Although the performance of both models on the same data is quite similar, the CNNGRU model is in some respects better, since it is less complex, having one less layer than the LSTM model, and also requiring a shorter training time than LSTM. Nevertheless, overall, the method developed in this article offers higher accuracy than the CNNGRU reported by Liu W. [32], mainly due to the specific selection of features and electrodes. They needed 16 electrodes to classify depressive people with an accuracy of 89,63%, while the current methodology reached almost perfect accuracy with a selection of only three electrodes and a few non-linear features. These methodological choices are crucial, as the CNNGRU model also improved its performance with our own data, using our selected  electrodes, features  and procedures.”

“The group of three electrodes selected in this study (FC2, AFz, F2) have been associated with  the activity of the Caudal Anterior Cingulate Cortex, according to the Desikan’s brain structural areas mapped on Giacometti’s electrodes mapping algorithm [2]. This is congruent with the literature reporting that the symptoms of depression are associated with anatomical and functional impairments in the Caudal Anterior Cingulate Cortex [35,36]. Note, however, that the anatomical parcelation methods, and the assignment of groups of electrodes to brain regions are primarily descriptive. To provide more accurate neuroanatomical and functional data on depression, neuroimaging methods or source estimation algorithms for high-density EEG set ups are necessary. The minimal approach of 3-electrodes EEG, used in this study, is appropriate for searching efficient machine learning classification algorithms, but not for accurate source estimation.”

“This study has some limitations, which may be addressed in future research. It was conducted with a population of young university students, half of whom showing depressive states, according to a self-report questionnaire, although none of them had been diagnosed with depression, and the results cannot be generalizable to other populations and depressive conditions.  However, given the simplicity of the EEG resting state protocol and the efficiency of the LSTM and CNNGRU as classificatory tools reported here, the current methodology could be applied in the future to broader and more heterogeneous samples of population, including groups of different ages and clinically diagnosed patients. The classificatory algorithms and method developed here provide objective measures of brain dynamics, which can be used as a complementary diagnosis tool in clinics. It could be trained to distinguish among different types of depression, depression levels and even anxiety and/or other comorbid disorders. In addition, it could be used for follow up, as an objective guidance of treatment results, that is applying the tool to depression patients before and after their treatment to test their improvement, beyond subjective impressions. Finally, this study raises the possibility of creating a portable EEG device with 3 electrodes, easy to use and even linked to smartphones, which could be used on primary care services to screen depressive cases, facilitating the prevention.”

10- The research findings and contribution need to be stated clearly. As well as the obtained results in this paper. So, the authors are requested to connect the main idea and contribution with the obtained results to prove that the proposed method accomplished its main aims and better results.

RESPONSE: Following the reviewer’s suggestion, we elaborated on these issues in the Discussion. Please, see response to point 9.

11- The logic of the introduction can be improved. For example, the reasons and significance of applying deep/machine learning methods to the neurobiology study of disorders could be introduced. Current progress and critical issues could also be mentioned. The authors can use these articles to edit this section. A Depression Diagnosis Method Based on the Hybrid Neural Network and Attention Mechanism-- Computer aided diagnosis system using deep convolutional neural networks for ADHD subtypes - A Depression Prediction Algorithm Based on Spatiotemporal Feature of EEG Signal- Detection of child depression using machine learning methods

RESPONSE: Thank you for revealing us very recent and relevant references. We mention and comment them in the introduction and discussion. As for the logic of using deep/machine learning methods to neurobiological signals, we think that the quest for objective measures, beyond self-report and clinical interview is a crucial argument developed in the introduction.

“Nowadays, depression diagnosis is mainly done through questionnaires and clinical interviews based on expertise of professionals with the guidance of WHO’s International Classification of Diseases (ICD 11) or DSM-V in the USA. Such an approach does not provide an entirely reliable diagnosis, since the patient’s answers rely on subjective wellness or expectancy to improve their feelings, besides possible emotional links between patients and doctors. All that can bias the interpretation of questionnaires, being necessary to advance in early depression detection through biomarkers that could help both on diagnosis and treatment evaluation [4,5].”

“The quest for objective diagnosis of mental disorders through non-invasive EEG has led to different approaches like the use of Evoked Potentials, Band Power, Signal Features, Functional Connectivity, Alpha Asymmetry, some of them with mean accuracies of 90% [6]. The combination of resting state data and Machine learning classification algorithms offers a promising new tool to improve the diagnosis and the evaluation of psychiatric disorders [7]. This combined method of resting state EEG and classificatory algorithms has also been applied to explore distinctive features of neural dynamics in depression [8]. Machine Learning (ML) consists of computer algorithms that can automatically improve their performance by being trained with available data sets. A new area of ML involves Deep Learning (DL) algorithms, which combine multi-layers of the same or different basic architectural modules producing successful goals like speech recognition, translation or even the creation of online chatbots that help people to improve mental health [8,9]. The growing field of applications based on robust Deep Learning has encouraged many researchers to use these methods with EEG resting state data classification [7], with special applications to explore psychopathological features and states, such as depression.”

 12- Please use the violin plot to show the distribution of extracted features in the two class of data.

RESPONSE:  In the new Figure 3, we offer the violin distribution of the extracted features both in the control and the depressive group.

Reviewer 2 Report

In this paper, a data driven approach was applied to select the most appropriate time window. However, there are certain aspects of the manuscript that still need improvements with major revision:

1、For the abstract part, the author should claim the background or pose the problem.

IMO, the references can not be appeared in this part.

2、For the introduction part, the authors should make a clear structure of the proposed method, e.g, background, problem, propose method.

3、For the manuscript, the structure should list as follows: introduction, related work, method, experiments, conclusion.

4、Some figures should be created from scratch. Colors should be standardized (in the whole paper), use one font for all things. Moreover, the quality of the images must be higher.

5、It is necessary for the authors to give the description of overfitting during training the model.

6、The author should provide the code link in the revised version.

7、The authors should compare the proposed methods with the state-of-the-art methods.

8、The authors should check the reference format, and keep the same format in the manuscript.

Author Response

REVIEWER 2

In this paper, a data driven approach was applied to select the most appropriate time window. However, there are certain aspects of the manuscript that still need improvements with major revision.

RESPONSE: Thank you for your revision and your comments and suggestions, which allowed us to improve the manuscript.

1For the abstract part, the author should claim the background or pose the problem.

RESPONSE: Thank you for your suggestion. The abstract has been rewritten to fulfil this requirement:

“Abstract: The growing number of depressive people and the overload in primary care services makes it  necessary to identify depressive states with easily accessible biomarkers such as mobile electroencephalography (EEG).  Some studies have addressed this issue by collecting and analyzing EEG Resting State in search of  appropriate features and classification methods. Traditionally, EEG resting state classification methods for depression were mainly based on linear or a combination of linear and non-linear features. We hypothesize that participants with ongoing depressive states differ from controls in complex patterns of brain dynamics that can be captured in EEG resting state data, using only nonlinear measures on a few electrodes,  making it possible to develop cheap and wearable devices that could be even monitorized through smartphones. To validate such a perspective, a resting state EEG study was conducted with 50 participants, half with depressive state (DEP) and half controls (CTL). A data driven approach was applied to select the most appropriate time window, and electrodes for the EEG analyses, as suggested by Giacometti [2]⁠, as well as the most efficient nonlinear features and classifiers, to distinguish between CTL and DEP participants. Nonlinear features showing temporo-spatial and spectral complexity were selected. The results confirmed that computing nonlinear features from a few selected electrodes in a 15-seconds time window are sufficient to classify DEP  and CTL participants accurately. Finally, after training and testing internally the classifier, the trained machine was applied to EEG resting state data (CTL and DEP) from a publicly available database, validating the capacity of generalization of the classifier with data from different equipment, population, and environment getting an accuracy near 100%.”

IMO, the references can not be appeared in this part.

RESPONSE: References were deleted from abstract.

2For the introduction part, the authors should make a clear structure of the proposed method, e.g, background, problem, propose method.

RESPONSE: Thank you for your request for clarification. The introduction has been partially rewritten to fulfil this requirement. So, at the end of the introduction we describe the method step by step:

“The current study aims to improve the classification of depression with nonlinear features, few electrodes and standardized classifiers, as well as the model recently suggested by Liu, et al. [32]. In this study, EEG resting state data were collected from a group of participants with depressive state and a group of matched controls, aiming to choose the best classification algorithm with a reduced number of electrodes and limited time-window, and test its generalization for an external sample. The design included four steps. First, a space-time boundary selection of the time window span and group of key electrodes, to use in further analyses. Second, identifying the key non-linear features and appropriate classifiers, which better capture the neural dynamics for the selected time-window and electrodes. Third, tuning, training and validating the selected classifier. The final model was trained with data collected in our laboratory, as described here. Fourth, the trained model was fed with external Resting State data from a publicly available database, validating the generalization capacity of the obtained model and the possibility of replication of our study.”

“In the first step, a logistic regression (LR) with a standard scaler (M=0, SD= 1) was used  as a baseline classifier. The use of basic non-linear features (Higuchi’s fractal dimension and Sample Entropy), suggested by Čukić, et al. [10], was chosen to explore the optimal time window span and group of electrodes. For such a purpose we used Giacometti's algorithm that links Desikan’s brain structural areas with 1020 standard EEG electrodes distribution [2].”

“In the second step, after having selected the time window and electrodes with first step results, different non-linear features were explored: 

  • Higuchi’s Fractal Dimension (Higuchi) is an approximate value for the box-counting dimension used on fractal analysis to graph a real-valued time series that has been used for more than 20 years in neurophysiological domains [13].
  • Spectral Entropy (SpectEnt) describes the complexity of a system based on applying the standard formula for entropy to the Power Spectral Density of EEG data. It quantifies the irregularity of EEG data [14].
  • Singular Value Decomposition Entropy (SVDEnt) characterizes information content or regularity of a signal depending on the number of vectors attributed to the process [15].
  • Sample Entropy (SampEnt) is a modification of approximate entropy, used for assessing the complexity of physiological time series signals [16].
  • Detrended Fluctuation Analysis (DFA) is a stochastic process, chaos theory and time series analysis. DFA is a method for determining the statistical self-affinity of a signal. It is useful for analyzing time series that appear to be long memory processes [17].
  • Permutation Entropy (PermEnt) gives a quantification measure of the complexity of a dynamic system by capturing the order relations between values of a time series and extracting a probability distribution of the ordinal patterns [34].”

“In the second step different classifiers were also tested:

  • Logistic Regression (LR): is a baseline dichotomic classifier from Machine Learning (ML)
  • Support Vector Machine (SVM): a robust ML algorithm that maps training examples to points in space (vectors) searching to maximize the distance  between categories
  • Multi Layer Perceptron (MLP): is one of the simplest Deep Learning (DL) models used in supervised learning, consisting of fully connected layers. 
  • Convolutional Neural Network (CNN): another DL algorithm mainly used in image data analysis due to their biological visual architecture similarity
  • Long Short Term Memory (LSTM): this DL algorithm is mainly used in sequential data analysis like natural language processing or time series analysis.
  • CNN+GRU (CNNGRU): this DL algorithm suggested by Liu W. [32] combines a CNN as a features extractor plus a Gated Recurrent Unit (GRU), an improvement of LSTM. “

“In the third step, after fixing the environments with previous steps results, the selected model was tuned, trained and validated with the data of our own experiment.”

“Finally, in the fourth step, the machine trained with our own dataset, resulting from such architecture and methodological approach was tested with external data from a different lab, population, and environment to verify the generality of this approach.”

3For the manuscript, the structure should list as follows: introduction, related work, method, experiments, conclusion.

RESPONSE: Thank you for your suggestions. The Introduction has been structured accordingly.

4Some figures should be created from scratch. Colors should be standardized (in the whole paper), use one font for all things. Moreover, the quality of the images must be higher.

RESPONSE: Thank you for your recommendation. All figures were recreated from scratch improved and saved at 600 dpi with more standard colors on the green-gray-blue scales.

5It is necessary for the authors to give the description of overfitting during training the model.

RESPONSE: Thank you for your important suggestion. Now, we include an updated Figure 7, showing the loss function values as the training/validating progress during training epochs, used to detect overfitting and underfitting. In underfitting cases the validation curve is above the training one, while in overfitting, the validation curve, after matching the training in some point, starts to grow up while the training curve keeps low.  In the Figure 7, we can see that the LSTM model validation curve converges to the highest accurately after 80 training epochs,  while the CNNGRU model both curves match each other so neither over nor under fitting affects the trained machines.

6The author should provide the code link in the revised version.

RESPONSE: Code link has been provided in the new version too.

7The authors should compare the proposed methods with the state-of-the-art methods.

RESPONSE: Thank you for your suggestion. We incorporated the latest CNNGRU model recently proposed by Liu W.  et al (2022) for depression classification and compared it with the standard models included LSTM. Please, see details in the updated sections “3.2.2. Testing and comparing different classifiers”, “3.3. Third step: tuning, training and validating with the experimental data”, and “3.4. Fourth step: validating the model with external data”, as well as in the updated Figures 6, 7 and 8, and . The current version of the manuscript shows that Liu’s approach even improves its performance with our choice of electrodes and nonlinear features, as explained in the  Discussion of the revised manuscript:

“Although the performance of both models on the same data is quite similar, the CNNGRU model is in some respects better, since it is less complex, having one less layer than the LSTM model, and also requiring a shorter training time than LSTM. Nevertheless, overall, the method developed in this article offers higher accuracy than the CNNGRU reported by Liu et al.,  [11], mainly due to the specific selection of features and electrodes. They needed 16 electrodes to classify depressive people with an accuracy of 89,63%, while the current methodology reached almost perfect accuracy with a selection of only three electrodes and a few non-linear features. These methodological choices are crucial, as the CNNGRU model also improved its performance with our own data, using our selected  electrodes, features  and procedures.”

8The authors should check the reference format, and keep the same format in the manuscript.

RESPONSE: All the references in the manuscript and the list of references were adapted to the Vancouver format, requested by the journal

Round 2

Reviewer 1 Report

The manuscript seems acceptable to me for publication in the journal with the corrections made. 

Reviewer 2 Report

The paper can be accepted after the minor revision.

1、The authors should improve the English in the revised manuscript.